# REVISITNG GRAPH NEURAL NETWORKS FOR TRAFFIC FORECASTING

## ABSTRACT

Accurate traffic forecasting is crucial for a wide range of traffic management applications. In recent years, Graph Neural Networks (GNNs) have emerged as one of the most promising methods to predict traffic. However, their complex architectures prevent them from being used in large networks and long-term forecasting. Although there are complex spatial-temporal dependencies in traffic data, the evolution of the traffic flow, in particular, is governed by linear dynamics, based on the law of flow conservation. Hence, we conjecture that linear models are sufficient for accurate traffic flow predictions. In this study, we investigate linear regression models to predict traffic flow. Models are created for different periods in the day, and exploit historical traffic data from the neighboring region as input. Using multiple real-world traffic data sets collected from the entire California highway systems, we demonstrate that our simple linear models outperform state-of-the-art GNNs by achieving both higher accuracy and significantly better efficiency. Moreover, we conduct comprehensive studies to analyze the impacts of various design elements of GNNs on the improvement of prediction accuracy. Based on our findings, we advocate re-considering the design of model architectures for traffic forecasting.

## 1 INTRODUCTION

Accurate traffic flow prediction is essential for various traffic applications, such as congestion prediction Wiering et al. (2004), traffic control Zhang & Masoud (2021), and traffic network design Fontaine & Minner (2014). However, due to the complexity of traffic dynamics, it is still an open research area. Traffic flow prediction approaches went through a progression from traditional statistical methods Ermagun & Levinson (2018) to machine learning models Sutskever et al. (2014). More recently, graph neural networks (GNNs) Li et al. (2017); Guo et al. (2019b); Lai et al. (2023); Zhong et al. (2023; 2022) have also been proposed for traffic forecasting because of the graph-based nature of traffic flow. These GNN models were shown to achieve the highest prediction accuracy. But their complex model architecture lead to significant computation and memory costs, limiting their applicability in large networks and long-term forecasting tasks. These barriers motivate us to develop a simpler model form that can be applied to large networks and produce predictions over long time horizons.

To this end, we revisit the dynamic characteristics of traffic flow, and design the most simple scheme that can sufficiently capture traffic flow patterns. As explained in Gerlough & Huber (1976), the dynamics in traffic flow must be constrained by the law of traffic flow conservation, which means that the change in traffic flow at a node should always be equal to the difference between the inflow and outflow of the vehicles at that node. We conjecture that this linear dependency between the past and future traffic flow states can be easily captured by a linear regression model. Hence, we investigate the efficacy of the linear regression scheme for traffic flow forecasting.

It should be noted that linear models do not have as much capacity as nonlinear models in handling high-dimensional inputs Worden & Green (2017). To tackle this challenge, we only select the traffic states of neighboring nodes as model input. Surprisingly, our results show that the proposed linear model outperforms most graph neural networks in terms of accuracy (up to 20% error reduction) and efficiency (improvement of two orders of magnitude of model training time). Based on detailed empirical studies of the model architectures and parameters, we find that, in contrast to the claims

in existing GNNs, the graph topology information may not be crucial in achieving more accurate traffic flow predictions. To sum up, the contributions of this work are listed below:

- To the best of our knowledge, this is the first work to challenge the effectiveness of GNNs for the task of traffic flow prediction.
- To validate our conjecture, we proposed a very simple linear regression scheme for traffic flow forecasting, which can be easily applied on large networks with significant reduction in model training time and increased accuracy.
- We conducted comprehensive analysis on the choice of model architecture and the effect that graph topology information, or lack thereof, has on the time series forecasting of flow over graphs with periodic patterns. Our findings will benefit future research in system state prediction of not only traffic system but also other infrastructure systems.

## 2 RELATED WORKS

Early data-driven approaches to traffic forecasting include methods such as Historical Average (HA) Klein (1997) and ARIMA Ermagun & Levinson (2018). In recent years, the field has seen a shift towards deep neural networks, such as recurrent neural network Madan & Mangipudi (2018) and long short-term memory networks Sutskever et al. (2014), which have demonstrated superior performance. These deep learning models, while powerful, often do not fully utilize the spatial dependencies within transportation networks.

Recognizing the importance of both spatial and temporal dependencies in traffic forecasting, recent approaches have combined Graph Neural Networks (GNNs) with time series modeling. The Diffusion-Convolution Recurrent Neural Network (DCRNN) Li et al. (2017) and Spatio-Temporal Graph Convolutional Networks (STGCN) Yu et al. (2017) are examples of models that incorporate spatial dependencies and temporal correlations. Adaptive Graph Convolutional Recurrent Network (AGCRN) Bai et al. (2020) introduced adaptive modules to capture node-specific patterns, while SCINet Liu et al. (2022) improved accuracy through sample convolution and interaction. Decomposition Dynamic Graph Convolutional Recurrent Network (DDGCRN) Weng et al. (2023) utilized dynamic graph convolution, and a Lightweight Framework for Correlated Time Series Forecasting (LightCTS) Lai et al. (2023) leveraged correlation information for efficiency and accuracy.

Recent advancements have also introduced attention-based architectures, such as Attention-Based Spatial-Temporal Graph Convolutional Networks (ASTGCN) Guo et al. (2019b), which incorporated a self-attention mechanism, and Propagation Delay-Aware Dynamic Long-Range Transformer (PDFormer) Jiang et al. (2023), which considered traffic delay-aware feature transformation. Spatio-Temporal Adaptive Embedding based Transformer (STAEformer) Liu et al. (2023) further improved performance using adaptive embedding techniques.

While existing models have excelled in short-term traffic forecasting (up to one hour), they face challenges in long-term traffic prediction and large (regional) traffic forecasting due to their computational complexity. The authors in James et al. (2021) proposed a novel graph neural network for long-term traffic prediction, leveraging manually curated historical data as the input sequence. This approach helps reduce the complexity associated with learning from long sequences, allowing for longer horizon predictions. In addition to addressing long-term forecasting, some studies have integrated DCRNN with graph partitioning techniques, as seen in DCR (2020). These techniques enable the training and prediction of traffic states in large networks. However, even with graph partitioning, training hundreds of models for partitioned subgraphs remains time-consuming. This study takes a different approach by suggesting the use of simple linear model form with an expert-based feature selection scheme for traffic forecasting, which is applicable to large network scenarios and long-term predictions.

## 3 METHODS

### 3.1 PROBLEM SETTING

The real-world traffic network topology is collected by OpenStreetMap Haklay & Weber (2008). The traffic road network is defined as a directed graph $G = (V, E, A)$, where $V$ is the set of $N = |V|$

nodes (i.e. traffic sensors), $E$ is the set of edges (sensors are connected when they are directly connected by a path without a sensor in between), and $A \in R^{N \times N}$ is the adjacency matrix that encodes the graph topology structure:

$$A_{i,j} = \left\{ \begin{array}{ll} 1, & \text{if } (v_i, v_j) \in E, \\ 0, & \text{otherwise,} \end{array} \right.$$

where $(v_i, v_j)$ represents the edge going from node $v_i$ to node $v_j$.

The traffic states (e.g. velocity, flow) at timestamp $t$ is denoted as $\boldsymbol{X}_t$. In traditional setting of traffic forecasting, we use historical traffic states collected over $K$ time steps, denoted by $\{\boldsymbol{X}_{t-K+1}, ..., \boldsymbol{X}_t\}$, to predict future traffic states over $Q$ time steps, denoted by $\{\boldsymbol{X}_{t+1}, ..., \boldsymbol{X}_{t+Q}\}$.

### 3.2 PREDICTION SCHEME

In this work, we build a prediction scheme based on linear regression for graph-structured time series forecasting. In this scheme, the traffic flow state at any future time step is simply a linear combination of historical traffic flow states. However, linear models do not have as strong capacity as nonlinear GNNs for dealing with high-dimensional input data. Therefore, we choose to pre-select the informative input features as inputs to improve the linear model performance.

We first consider that the traffic state of one node will be only affected by traffic movement in nearby locations. We define $\mathcal{N}_i^H$ as the set of nodes that are at most $H$ hops away from the center node $v_i$, where $H$ is a hyper-parameter. Hence, the prediction of traffic flow at node $v_i$ at the $p$-th future time step is:

$$\boldsymbol{x}_{t+q}^i = \sum_{j \in \mathcal{N}_i^H} b_{j,q} \boldsymbol{x}_t^j,$$

where $\boldsymbol{x}_t^i$ is the traffic flow states of node $v_i$ at time step $t$, and $b_{j,q}$ is the trainable parameter of the linear model for the future $q$-th time step traffic flow states.

Moreover, most real-world time series data, such as traffic and electricity consumption data, show strong daily periodic patterns. Therefore, we also train different linear models for different periods within a day, e.g. 9am-10am and 9pm-10pm. To do so, we divide each day into periods of equal length $\tau$, and train separate linear models for each period. Under this setting, our prediction scheme is represented as:

$$\boldsymbol{x}_{t+q}^i = \sum_{j \in \mathcal{N}_i^H} b_{j,q}^l \boldsymbol{x}_t^j, \quad t \in P_l,$$

where $b_{j,q}^l$ represents the weight of historical traffic states $\boldsymbol{X}_t^j$ for the $p$-th time step future prediction, when the current time step falls in the $l$-th period.

We highlight the differences in prediction scheme between GNNs and our study in Figure 1. GNNs take multiple snapshots of historical traffic data as input and predict multiple snapshots of the future traffic data simultaneously. In contrast, our method only uses the latest traffic snapshot as input and trains different linear models to predict traffic states at different times in future. In addition, our method aggregates the neighbor node information in a much smaller region compared to GNNs, which use stacked graph convolution layers. Despite these variations in our prediction scheme, it's important to note that we utilize less historical traffic information compared to GNNs.

### 3.3 MODEL TRAINING

The trainable parameters of our linear models are determined by closed-form solution instead of gradient descent method. Since the traffic dynamics is modeled as a discrete-time linear dynamic system, we can represent our linear regression model as:

$$\boldsymbol{X}_{t+q}^i = \boldsymbol{X}_t^{\mathcal{N}_i^H} B_q^l, \quad t \in P_l,$$

where $\boldsymbol{X}_{t+q}^i \in \mathcal{R}^M$ is the traffic state of node $v_i$ and $M$ is the number of recorded states in $P_l$, $\boldsymbol{X}_t^{\mathcal{N}_i^H} \in \mathcal{R}^{C \times M}$ is the traffic state of neighbors of node $v_i$ and $C$ is the number of nodes in $\mathcal{N}_i^H$.

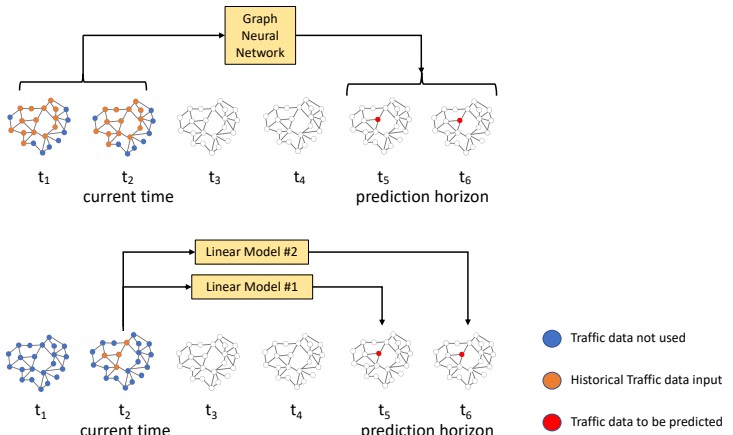

Figure 1: The comparison between the Prediction scheme used in our study and the prediction scheme used in most GNN models is shown. We compare the accuracy and efficiency (training and inference time) between one GNN model forecasting traffic states of future $Q$ time steps, and $Q$ distinct linear models predicting the traffic states of each of those future time steps.

We can estimate $B_q^l$ with the closed form formula:

$$B_q^l = \{\boldsymbol{X}_t^{\mathcal{N}_i^H}\}^{\ddagger} \boldsymbol{X}_{t+q}^i,$$

where $\ddagger$ represents the pesudo-inverse of the matrix. Without using gradient descent to estimate the parameters, we can train our linear models in a much more efficient way.

## 4 EXPERIMENTS

### 4.1 EXPERIMENTAL SETUP

We begin by evaluating the performance of our proposed model (i.e. Linear) on multi-step forecasting tasks using two widely-used benchmark datasets, PEMS04 Guo et al. (2019a) and PEMS08 Guo et al. (2019a), which were collected from two regions of California's highway network. In addition, we also report results for the entire highway system of California in Subsection 4.3. The traffic flow was measured in vehicles per hour. To establish baselines for comparison, we utilized the following five machine learning models:

- Long short-term memory (LSTM) Sutskever et al. (2014): a popular RNN-based sequence model for multiple time series forecasting tasks.
- Diffusion convolutional recurrent neural network (DCRNN) Li et al. (2017): the first graph neural network model for traffic forecasting.
- Attention based spatial temporal graph convolutional networks (ASTGCN) Guo et al. (2019b): the first attention mechanism based graph neural network for traffic forecasting.
- LightCTS Lai et al. (2023): A Lightweight Framework for Correlated Time Series Forecasting based on linear correlation analysis.
- STAEformer Liu et al. (2023): A transformer-based method that currently has the best reported performance on traffic flow forecasting.
- Koopman operator (KO) Avila & Mezić (2020): A linear data-driven method for highway traffic forecasting using Koopman theory.

The baseline models were trained using the same settings as in their original papers on a Quad A40 GPU. The hyper-parameter search of Linear models was selected based on the grid search Liashchynskyi & Liashchynskyi (2019). The prediction accuracy of different models was quantified using mean absolute error (MAE). Model lightness and memory complexity were evaluated based

on the number of parameters and peak memory usage. The computational complexity was also reported by measuring the time for model training and inference.

## 4.2 BENCHMARK TRAFFIC DATA SETS FROM CALIFORNIA DISTRICTS

In Table 1, we present the performance of our models for different prediction horizon lengths. While most previous works evaluated their models using a prediction horizon of only 60 minutes, it is important to consider that traffic congestion can often last for hours as urban populations increase James et al. (2021). Therefore, in our study, we extended the evaluation to include predictions up to four hours ahead.

Table 1: The prediction accuracy of different models is shown. The lowest prediction error is marked as bold and the second lowest prediction error is marked with underscore.

| Mean Absolute Error (unit: vehicle / hour) | | | | | | |
|---|---|---|---|---|---|---|
| Prediction horizon | Model | PEMS04 | | | PEMS08 | | |
| | | 15min | 30min | 60min | 15min | 30min | 60min |
| 60min | LSTM | 21.85 | 25.91 | 34.80 | 17.61 | 21.10 | 28.98 |
| | DCRNN | 20.37 | 23.62 | 30.49 | 15.48 | 17.65 | 21.76 |
| | ASTGCN | 19.20 | 20.58 | 23.57 | 16.19 | 18.12 | 22.12 |
| | LightCTS | 19.01 | 20.69 | 22.41 | 14.28 | 15.43 | 16.72 |
| | STAEformer | **18.79** | **19.98** | 22.12 | **13.75** | **14.61** | **16.20** |
| | KO | 20.82 | 21.52 | 26.09 | 18.57 | 20.64 | 22.04 |
| | Linear (Ours) | 20.09 | 20.47 | **21.95** | 14.67 | 15.32 | 16.67 |
| 120min | | 60min | 90min | 120min | 60min | 90min | 120min |
| | LSTM | 37.82 | 39.21 | 41.37 | 33.88 | 36.31 | 38.82 |
| | DCRNN | 34.23 | 37.53 | 38.91 | 24.71 | 27.03 | 29.64 |
| | ASTGCN | 25.27 | 27.31 | 28.03 | 24.36 | 26.15 | 27.42 |
| | LightCTS | 23.11 | 23.84 | 24.47 | 17.83 | 18.42 | 19.07 |
| | STAEformer | 22.42 | 23.62 | 24.24 | 17.25 | 18.05 | 18.58 |
| | KO | 26.09 | 27.53 | 28.32 | 22.04 | 23.51 | 24.94 |
| | Linear (Ours) | **21.95** | **22.95** | **23.75** | **16.97** | **17.85** | **18.38** |
| 240min | | 120min | 180min | 240min | 120min | 180min | 240min |
| | LSTM | 44.72 | 46.80 | 48.25 | 41.67 | 43.69 | 47.84 |
| | DCRNN | 41.26 | 43.15 | 45.49 | 32.14 | 34.85 | 36.59 |
| | ASTGCN | 30.71 | 32.47 | 35.01 | 29.07 | 31.55 | 33.19 |
| | LightCTS | 28.21 | 30.03 | 32.84 | 20.55 | 23.11 | 26.03 |
| | STAEformer | 27.97 | 29.78 | 32.03 | 20.24 | 22.72 | 25.45 |
| | KO | 28.32 | 29.75 | 31.53 | 24.94 | 25.85 | 26.20 |
| | Linear (Ours) | **23.75** | **24.89** | **25.96** | **18.44** | **19.45** | **19.99** |

We first conducted an evaluation of various models for predicting future 60-minute traffic states, including predictions at 15-minute, 30-minute, and 60-minute ahead time intervals. The Koopman operator achieved higher accuracy than LSTM on both datasets, highlighting the effectiveness of a linear model in capturing traffic dynamics when the graph topology is not taken into consideration. Furthermore, in around two-thirds of the cases, our proposed Linear model achieved the best prediction accuracy, while in the remaining cases, our performance was very close (off by only 1 vehicle per hour) to the best performance (offered by a graph neural network model). Overall, it can be seen that as the prediction horizons is increased, the advantage that our model offers over existing methods increases. To better visualize the model prediction, we show the 240-min ahead predicted time series and ground truth time series in Figure 2. We assert that our model adequately captures the traffic patterns for both roads with high traffic volume and those with low traffic volume.

Table 2 compares the computational efficiency of the studied methods. Our proposed Linear model achieved a significant reduction of the number of model parameters (over 99%). Furthermore, we observed that linear models had much smaller model complexity than deep learning models, requiring only 20% of the peak memory usage of model inference of the lightest deep learning model (i.e., LightCTS). The number of parameters in Linear models was less than 5% of the number of parameters in LightCTS, and the peak memory usage of Linear models was less than 1% of the peak memory usage of LightCTS.

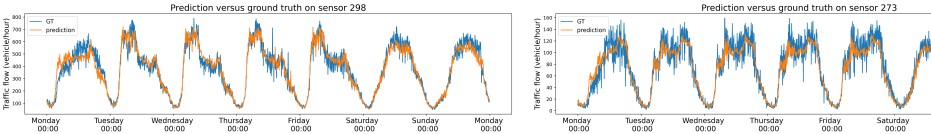

(a) Prediction performance on high traffic flow link     (b) Prediction performance on low traffic flow link

Figure 2: The 240-min ahead predicted time series and ground truth time series of two locations of one week are shown in the figure. The left one shows the model performance on high-traffic-flow link and the right one shows the model performance on low-traffic-flow link.

Our proposed method also achieved significant improvements in model training time, taking only about 2.5% of the LightCTS training time to estimate the Linear models. This is supported by the tabulated inference times, highlighting the computational efficiency of our proposed method.

Table 2: The efficiency comparison between different models is shown. The highest efficiency indicator is marked as bold and the second highest efficiency indicator is marked with underscore.

| Dataset | Model | Params (K) | Memory (Mb) | Training time (hour) | Inference time (second) |
|---------|-------|-----------|-------------|----------------------|-------------------------|
| PEMS04 | LSTM | 124 | 6.5 | 2.0 | 54 |
| | DCRNN | 371 | 8.1 | 9.0 | 110 |
| | ASTGCN | 254 | 7.2 | 2.5 | 13 |
| | LightCTS | 185 | 4.7 | 1.1 | 4.8 |
| | STAEformer | 267 | 7.6 | 3.1 | 15 |
| | KO | 1130 | 0.75 | 0.11 | 3.2 |
| | Linear (Ours) | **7.2** | **0.031** | **0.021** | **0.35** |
| PEMS08 | LSTM | 122 | 3.7 | 1.1 | 25 |
| | DCRNN | 371 | 4.8 | 4.5 | 67 |
| | ASTGCN | 248 | 3.2 | 1.3 | 9.4 |
| | LightCTS | 177 | 2.8 | 0.43 | 2.6 |
| | STAEformer | 257 | 3.6 | 1.5 | 11.1 |
| | KO | 346 | 0.23 | 0.06 | 1.8 |
| | Linear (Ours) | **3.8** | **0.026** | **0.016** | **0.18** |

## 4.3 California highway system

We further evaluated our approach using the traffic data set collected from the entire California highway network DCR (2020), which includes real-time traffic features collected from over 37,000 individual sensors. Due to the high computational cost of training the model on this large data set, we only compared the performance of our model with that of LightCTS, which was trained on GPUs with a 40GB memory limit. With this memory capacity, LightCTS can only load the adjacency matrix of a network with up to 10,000 nodes. Therefore, in addition to the entire network, we also considered two cases where subgraphs of the entire network were considered.

Table 3 shows the prediction errors for the three graph sizes. For the training of LightCTS, we had to set the batch size to be 32 for the 1,000-node subgraph and 1 for the 10,000-node subgraph in order to satisfy the GPU memory constraint. We trained and evaluated our model and the LightCTS on these subgraphs for different prediction horizons (60min, 120min, 180min, and 240min). The reported error is the average prediction error over the entire prediction horizon. It can be seen that our model achieved better performance compared to the state-of-the-art LightCTS model. We observe that LightCTS has a higher performance drop than our model when trained on a larger network. This is because the performance of machine learning models is dependent on the batch size used during training, and achieving satisfactory training convergence with smaller batch sizes can be challenging.

In addition to prediction accuracy, we report additional information on the computation complexity and memory complexity for the large California highway network data set in Table 4. In particular, the table compares our proposed Linear models with LightCTS in terms of peak memory and time

Table 3: The prediction accuracy comparison between graph Koopman and LightCTS is shown. The lower prediction error is marked as bold.

| Graph Size | Model | Predicted time stamp | | | |
|---|---|---|---|---|---|
| | | 60min | 120min | 180min | 240min |
| 1,000 | LightCTS | **18.41** | 21.32 | 23.75 | 27.19 |
| | Linear (Ours) | 18.76 | **19.93** | **21.54** | **22.03** |
| 10,000 | LightCTS | 20.25 | 22.17 | 24.29 | 26.89 |
| | Linear (Ours) | **18.92** | **20.01** | **21.78** | **22.42** |
| 37,000 | Linear (Ours) | 19.32 | 20.21 | 22.08 | 22.72 |

used during model training. The batch size for LightCTS was set to be one for a fair comparison across cases of different graph sizes. It can be seen that our model is significantly more computation-efficient and memory-efficient than LightCTS. For instance, while LightCTS could only handle traffic networks of up to 10,000 nodes with a 40GB memory GPU, our model can be trained on the entire large network using less than 2GB memory. Our model also requires significantly lower peak memory usage during model training compared to LightCTS. Additionally, our model needs much shorter training time compared to LightCTS, with the efficiency improvement being about one order of magnitude for the graph of 10,000 nodes.

Table 4: The efficiency comparison between Linear models and LightCTS is shown. The higher efficiency indicator is marked as bold.

| | Model | Graph size | | |
|---|---|---|---|---|
| | | 1000 | 10000 | 37000 |
| Training time | LightCTS | 1.8 | 46 | / |
| (unit: hour) | Linear (Ours) | **0.031** | **0.642** | **1.739** |
| Training peak memory | LightCTS | 0.73 | 38.9 | / |
| (unit: GB) | Linear (Ours) | **0.325** | **1.245** | **1.932** |

## 4.4 ABLATION STUDIES

We also conducted a hyper-parameter analysis for the 60min prediction horizon setting. In particular, we varied the period length $\tau$ and the number of hops $H$ (determining the neighborhood size) and evaluated the corresponding prediction errors, as shown in Figure 3. It can be seen that our model is not sensitive to the choice of period length and that smaller values of $h$ are generally better for performance.

We found that when the model considered a large region of neighbors (i.e., larger $h$), it required more data to accurately calculate the Linear models. Therefore, if the hop value is large (i.e., $h > 1$), our model may suffer from a decrease in prediction accuracy with a smaller period length. It can be seen that for these data sets, the best performance is achieved when $h = 0$. This indicates that the traffic prediction at one location has strong linear dependency on its own historical traffic data, and none on information from neighboring sensors, which is consistent with the conclusion reported Zhong et al. (2022).

## 5 DISCUSSION

To further explain and analyze the main results in this work, we implemented additional experiments measuring how the performance change when (1) the nonlinearity level of the model is changed, (2) the graph topology information is exploited in the model, and (3) different traffic state variables are predicted.

### 5.1 DOES MODEL COMPLEXITY ALWAYS HELP IN TRAFFIC FLOW FORECASTING?

In most cases in machine learning domains, deep learning architectures with higher nonlinearity are usually considered to better capture complex system dynamics. However, in this work we showed that linear models could outperformed all the baseline GNNs. To better investigate these results, we

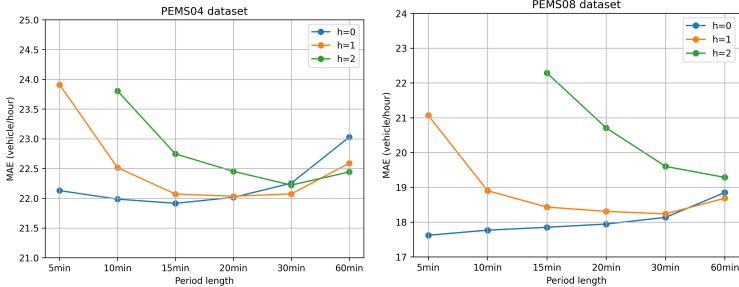

Figure 3: The 60-min ahead prediction accuracy of Linear models with different hyper-parameters is shown. The model performance for PEMS04 data set is shown on the left and the model performance for PEMS08 data set is shown on the right.

evaluated the accuracy of multi-layer perceptrons (MLP) with different numbers of layers for traffic flow forecasting.

In particular, in the MLP models, we used Tanh Karlik & Olgac (2011) function as the activation function. The average prediction errors over the whole prediction horizon of different models are shown in Table 5. As the number of layers in MLP increases, the prediction accuracy drops, which shows that higher model complexity is not always beneficial.

Table 5: The accuracy comparison between multiple MLP models and our method is shown. The higher accuracy indicator is marked as bold.

| Mean Absolute Error (unit: vehicle / hour) | | | | | | |
|---|---|---|---|---|---|---|
| Model | PEMS04 | | | PEMS08 | | |
| | 60min | 120min | 240min | 60min | 120min | 240min |
| MLP (1 layer) | **22.15** | **24.87** | **29.12** | **17.25** | **18.88** | **21.75** |
| MLP (2 layers) | 24.15 | 27.42 | 30.82 | 19.54 | 21.72 | 24.91 |
| MLP (3 layers) | 24.12 | 27.75 | 30.95 | 19.12 | 22.98 | 24.48 |

## 5.2 IS GRAPH TOPOLOGY INFORMATION IMPORTANT FOR TRAFFIC FLOW FORECASTING?

In our ablation studies, we observed in Figure 3 that our model performs best when traffic information from neighboring nodes is not taken into consideration. This was in contrast to the main motivation behind using GNNs for traffic forecasting. To further analyze the effects that neighboring nodes have, we studied the effect of removing adjacency information from GNN architectures. In particular, we replaced the original adjacency matrix with the identity matrix, and used the same model architecture. Table 6 shows the changes in prediction errors due to the removal of adjacency information. Surprisingly, we observed that this substantial change in graph topology information only had limited effects on the model performances. In some cases, using removing the adjacency information even led to better model performances. These results ask for a more thorough study into when graph topology information is useful in traffic forecasting.

## 5.3 DOES CONSERVATION LAW AFFECT THE PREDICTIONS?

The main results shown in Seciton 4 are for traffic flows. Another traffic variable that is often used in network modeling is the speed. Speed dynamics is nonlinear and more difficult to predict compared to traffic flow. To further investigate the promise of our proposed linear model, we also evaluated its performance on traffic speed prediction. Figure 7 compares the performance of our model versus DCRNN and STAEformer, when evaluated on PEMS-bay Li et al. (2017) and Metr-LA Li et al. (2017) data sets. It can be seen that our model didn't achieve better results compared to GNN models. This shows that the advantage of our model depends on the choice of traffic variable. In particular, when traffic flow, which is governed by linear law of conservation, is predicted, our linear model can be particularly advantageous.

Table 6: Comparison between prediction errors for GNN models using the adjacency matrix (before the arrows), and GNN models using no adjacency information (after the arrows). The smaller errors are shown in boldface.

| Mean Absolute Error (unit: vehicle / hour) | | |
|---|---|---|
| Model | PEMS04 | |
| | 60min | 120min | 240min |
| lightCTS | $20.52 \rightarrow 20.89$ | $23.91 \rightarrow 23.11$ | $28.55 \rightarrow 28.64$ |
| STAEformer | $20.22 \rightarrow 20.45$ | $23.37 \rightarrow 22.98$ | $28.12 \rightarrow 27.64$ |
| DCRNN | $22.43 \rightarrow 23.61$ | $25.17 \rightarrow 26.39$ | $29.82 \rightarrow 28.45$ |
| ASTGCN | $27.95 \rightarrow 27.88$ | $29.02 \rightarrow 28.54$ | $31.82 \rightarrow 29.99$ |
| Model | PEMS08 | |
| | 60min | 120min | 240min |
| LightCTS | $15.51 \rightarrow 15.93$ | $18.22 \rightarrow 18.37$ | $21.45 \rightarrow 20.93$ |
| STAEformer | $15.39 \rightarrow 15.73$ | $17.98 \rightarrow 17.14$ | $21.05 \rightarrow 20.88$ |
| DCRNN | $17.91 \rightarrow 18.77$ | $19.12 \rightarrow 20.11$ | $22.15 \rightarrow 22.03$ |
| ASTGCN | $21.54 \rightarrow 20.89$ | $23.72 \rightarrow 23.41$ | $25.91 \rightarrow 25.78$ |

Table 7: The accuracy comparison between our method and state-of-art models. The higher accuracy indicator is marked as bold. $\delta$ is denoted as the missing ratio of data points.

| | Mean Absolute Error (unit: miles / hour) | | | | | |
|---|---|---|---|---|---|---|
| Model | PEMS-bay ( $\delta$=0.003%) | | | Metr-LA ($\delta$=8.11%) | | |
| | 60min | 120min | 240min | 60min | 120min | 240min |
| DCRNN | 2.07 | 2.31 | 2.67 | 3.60 | 4.21 | 6.53 |
| STAEformer | **1.89** | **2.22** | 2.64 | **3.42** | **3.99** | **5.68** |
| Linear (Ours) | 2.21 | 2.43 | **2.54** | 3.94 | 4.46 | 5.74 |

## 6    CONCLUSION AND LIMITATION

In summary, this paper proposed to use extremely simple linear models for capturing the flow dynamics over a graph. Surprisingly, the proposed model outperforms all existing graph neural networks for traffic flow forecasting tasks, demonstrating the potential of linear models in capturing nonlinear graph-based dynamics, when the targeted features follow conservation laws (e.g. water volume, electric current, air flow volume). The Linear models has few hyper-parameters, making it easy to implement and tune for practical applications on large networks and long-term forecasting. Our findings out of the ablation studies is also beneficial for future research in system state prediction of not only traffic system but also many other infrastructure systems with periodic patterns. We also conducted a thorough examination of the performance of state-of-the-art models. Our findings show the nonlinearity and the graph topology is not always beneficial for capturing traffic dynamics. Hence, we recommend a more comprehensive investigation into the impact of graph convolution operations on tasks related to traffic forecasting.

Despite the promising results of our proposed method, there are still some limitations that need to be addressed. Our method is built upon on the law of traffic flow conservation, and only showed significant superiority in traffic flow forecasting. To adopt the linear models for other traffic features (e.g. traffic speed), we can combine with symbolic regression to uncover the hidden relationship between traffic flow and traffic speed. Also, its performance on aperiodic or more general dynamics needs to be investigated. In future works, we plan to also investigate the capability of our model to manage high-dimensional inputs and explore the application of the Linear models in other domains.

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
