# OpenReview forum: "Revisitng graph neural networks for traffic forecasting"
_ICLR.cc/2024/Conference — ICLR 2024 Conference Withdrawn Submission_

### Official Review · Reviewer_gedi · 2023-10-21

**Soundness:** 2 fair
**Presentation:** 3 good
**Contribution:** 2 fair
**Rating:** 3
**Confidence:** 4

**Summary:**

The paper investigates linear regression models to predict traffic flow, in particular as an alternative to computationally costly graph neural networks. The paper finds the presented linear method to be more accurate, and dramatically more efficient to run.

A key assumption that the authors make is that traffic flow conserves flow, so that its dynamics can be modeled using linear methods.

**Strengths:**

- The paper investigates traffic flow modeling, a question of significant practical importance.
- The paper pays attention to whether models can be applied at scale.
- The paper finds that the proposed method achieves state-of-the-art performance, beating a number of significantly more sophisticated methods (but see “Weaknesses” below).

**Weaknesses:**

- The paper’s scientific contribution is not significant enough to warrant publication at ICLR. The paper uses a well-known and commonly used method (linear regression) in an application where the same method may have been used before (but not published). The paper may be more suited for a conference or journal that is dedicated to traffic modeling.
- Even though this paper is empirical in nature, the authors do not release any code. Moreover, the descriptions of experiments are insufficient in detail to reproduce the experimental results.

**Questions:**

- The word “Revisiting” in the paper’s title is misspelled (“Revisitng”)

---

### Official Review · Reviewer_NDtY · 2023-10-30

**Soundness:** 2 fair
**Presentation:** 3 good
**Contribution:** 1 poor
**Rating:** 5
**Confidence:** 4

**Summary:**

The article introduces a lightweight linear model that demonstrates strong performance in long-term spatiotemporal sequence prediction tasks, prompting a reevaluation of the necessity to employ GNNs or attention mechanisms for capturing spatial dependencies. This encourages a reconsideration of the effectiveness of complex models employed in the past.

**Strengths:**

1.The proposed model in the article is remarkably elegant and simple, as it doesn't require gradient descent but instead obtains model parameters through closed-form solutions of equations. This results in very low computational requirements.
2.The model has an exceptionally low parameter count, consuming minimal GPU memory, and exhibits potential for scaling to large graphs.
The authors have conducted thorough experiments, showcasing the model's strong performance in long-term prediction tasks across multiple datasets, outperforming many complex models.
3.The article offers a detailed analysis of model complexity and graph topology, supported by empirical results that highlight the excellent performance of this concise model.

**Weaknesses:**

1.As mentioned in the article, the model's performance is less effective in predicting traffic variables such as vehicle speed compared to GNN models.
2.In the most common spatiotemporal prediction task where one-hour historical data is used to predict one-hour future traffic flow, the linear model proposed in the article did not perform well. It lagged behind many GNN and attention-based models. This raises questions about whether the superiority of the linear model in long-term prediction tasks over previous models is due to a lack of thorough hyperparameter tuning in the comparison.

**Questions:**

see my concerns

---

### Official Review · Reviewer_R1zg · 2023-10-30

**Soundness:** 2 fair
**Presentation:** 3 good
**Contribution:** 2 fair
**Rating:** 3
**Confidence:** 5

**Summary:**

This paper proposes a traffic prediction model based on linear regression scheme rather than graph neural network. The authors claim that this is the first work to challenge the effectiveness of GNNs for traffic prediction and the authors believe the proposed simple model achieve SOTA prediction performance on several widely used traffic prediction datasets. The idea of graph-free traffic prediction model has been studied and the reported result of the proposed model is not that good than SOTA baselines.

**Strengths:**

1. It is appreciated to challenge the effectiveness of GNNs for traffic prediction.
2. The idea of using linear layers to replace GNNs is interesting.
3. The time and space consumption of the proposed model is much better than existing works.

**Weaknesses:**

1. The idea of graph-free traffic prediction is not new, e.g., [1]. And this work actually utilizes the graph structures as the neighboring nodes are used.
2. Figure 1 does not provide enough information about model architecture, making the proposed framework not clear enough.
3. The reported result is problematic. The reported performance of STAEformer in this paper is far worse than that in [2]. Even so, the performance of proposed model still struggles to surpass that of baselines in the most commonly used setting, i.e., prediction horizon as 60min.

[1] Liu X, Liang Y, Huang C, et al. Do We Really Need Graph Neural Networks for Traffic Forecasting?[J]. arXiv preprint arXiv:2301.12603, 2023.
[2] Liu H, Dong Z, Jiang R, et al. Spatio-temporal adaptive embedding makes vanilla transformer sota for traffic forecasting[C]//Proceedings of the 32nd ACM International Conference on Information and Knowledge Management. 2023: 4125-4129.

**Questions:**

It would be appreciated if the authors could provide the code.

---

### Official Review · Reviewer_d1Pk · 2023-10-31

**Soundness:** 3 good
**Presentation:** 3 good
**Contribution:** 3 good
**Rating:** 5
**Confidence:** 3

**Summary:**

The authors investigated linear regression models to predict traffic flow. Models are created for different periods in the
day, and exploit historical traffic data from the neighboring region as input. Using real-world traffic data sets collected from the entire California highway systems, the authors demonstrated that the simple linear models outperform state-of-the-art GNNs by achieving both higher accuracy and significantly better efficiency. They also conducted studies to analyze the impacts of various
design elements of GNNs on the improvement of prediction accuracy. The results show the nonlinearity and the graph topology are not always beneficial for capturing traffic dynamics. The authors recommend a more comprehensive investigation into the impact of graph convolution operations on tasks related to traffic forecasting. The method is built upon on the law of traffic flow conservation and only showed significant superiority in traffic flow forecasting.

**Strengths:**

- Non-mainstream idea to study a simple linear regression model for traffic forecasting. It is interesting that the results are so good compared to the approaches based on neural networks.
- I like the ablation study.
- There is a good discussion of limitations.
- The paper is well-written, I found only some minor writing issues.
- Comparison of not only accuracy/errors of prediction but also time of training and inference
- The authors claim that this is the first work to challenge the effectiveness of GNNs, so despite the simplicity of the proposed method, the impact can be significant.

**Weaknesses:**

- The title is misleading because the main contribution is based on applying a simple linear regression model to traffic prediction. On the other hand, the authors advocate reconsidering the design of graph neural network architectures for traffic forecasting.
-  The originality is based on challenging the effectiveness of GNNs, there is no innovation from the methodological perspective. However, it seems that the challenge might be successful.
- I don't see information about the availability of the code, so it is not possible to verify or reproduce the results.
- Some writing issues:
   - p. 3: P_l is not explained
   - p. 3: p-th time step -> q-th time step
   - p. 4: "Prediction scheme" -> "prediction scheme"
   - p. 7: "could outperformed" -> "could outperform" or "outperformed"
   - p. 8: table 5 does not contain comparison with the introduced linear model, but in the caption there is a sentence "The accuracy
     comparison between multiple MLP models and our method is shown."
   - p. 9: "nonlinearity and the graph topology is not always" -> "nonlinearity and the graph topology are not always"

**Questions:**

-